# Application Development of Smoke Leakage Test Apparatus for Door Sets in the Field

**Hsuan-Yu Hung [1],\*, Ching-Yuan Lin [1], Ying-Ji Chuang [1] and Chung-Pi Luan [2]**

[1] Department of Architecture, National Taiwan University of Science and Technology, Taipei 10607, Taiwan; linyuan@mail.ntust.edu.tw (C.-Y.L.); chuang@mail.ntust.edu.tw (Y.-J.C.)

[2] Construction and Planning Agency, Ministry of The Interior, Taipei 10556, Taiwan; pierre@cpami.gov.tw

\* Correspondence: D10913008@mail.ntust.edu.tw or auuv2000@gmail.com

**Abstract:** Heavy smoke from building fires is the main cause of casualties. As smoke typically diffuses through building openings, smoke control performance of building openings is critical to survival and requires considerable attention. In the past, the detection method could only be used in the laboratory, and the detection equipment could not be moved. Therefore, the main purpose of this research was to develop a methodology for field testing of smoke control properties of doors in order to ensure that the smoke control performance of doors tested in the laboratory and doors installed in the field can be realized without any discrepancy. Furthermore, this test method underwent a comparison test with the CNS 15038 "Method of Test for Evaluating Smoke Control Performance of Doors" for the same subject. The test results showed no significant difference based on independent sample testing, demonstrating the feasibility of this test method and test apparatus. The instrument developed by this research is light and easy to carry, and the operation method is simple. Such a test method can be applied to different doors and is non-destructive, non-hazardous, and reusable. In the future, by extending the design principle of the system, this test method can be applied to other fire protection equipment for the inspection of smoke control capabilities and can be used as a reference for relevant organizations to establish test specifications and standards.

**Keywords:** door; smoke; pressure difference; leakage test; building security

## 1. Introduction

According to previous research regarding building fires, the main factor contributing to human casualties is the dispersion of smoke through building openings [1]. The mortality rate due to this is over 50% greater than the mortality rate due to causes directly related to fires [2], which is why the smoke control performance of building openings has particular importance. Currently, the smoke control performance of building doors has been specified in various countries [3–9], requiring qualified smoke control doors to provide smoke control capabilities under both medium-temperature and ambient-temperature conditions. As a consequence of the complexity of fire scenarios, ambient-temperature smoke may also be life threatening [10]. For example, if a source of fire is relatively distant from a door, the temperature of the smoke near the door may be relatively low or even comparable to ambient temperatures. In such a scenario, while the door may not be damaged by the heat of the fire, individuals on the rear side of the door may be fatally exposed to excessive smoke inhalation. Therefore, a door's fire protection effectiveness may not be the primary concern. Instead, what is necessary is the door's capability of resisting against smoke inflow at ambient temperatures and under certain pressures. One example is BS 476-31 [11], which only requires smoke control performance at ambient temperature conditions. Doors with better smoke control capabilities at ambient temperatures can also prevent other toxic gases from entering the interior space under different circumstances, reducing the occurrence of related disasters.

According to Chuang et al. [12,13], Wu et al. [14], Tsai et al. [15], Gross [16], and Kuo et al. [17], under the same differential pressure conditions, when the temperature of the air increases, the volume increases and its density relatively decreases. As a result, the volume leakage of the air through the gap of the door will be reduced. In other words, if the volume of air leakage through the test body gap at ambient temperatures is available, identification whether the smoke control performance of the test subject at medium temperatures can meet the requirements of national regulations can be achieved, provided there is no deformation or damage of the material at medium temperatures. However, the components of doors for architectural use not only consist of steel, but also wood and organic materials, aluminum alloys, or plastics, as well as the use of filler strips, expansion materials, and glass windows. In 2000, Rakic [18] studied the effects of the presence and absence of filler strips on the smoke control performance of doors by means of UL 1784 [9]. The conclusion is that the installation of filler strips in doors could effectively increase their smoke control performance, demonstrating that filler strips and the methods used for the construction of door seams are the key to smoke control performance. However, these materials may be affected by thermal and physical or chemical transformations at medium temperatures, which may lead to inflation, flaming, or glass breakage. Therefore, medium-temperature testing is essential as this step of testing reveals the conditions of heat-induced deformation and damage to materials. To pass the test, the materials must possess capability of withstanding medium temperatures. Therefore, the premise of this field test method was that the test door should first pass the CNS 15038 [3] or ISO 5925-1 test [5] in the laboratory. The field test should only be conducted at ambient temperatures to determine whether the materials and methods applied can meet practical smoke control performance, so that the smoke control performance of doors tested in the laboratory and doors installed in the field can be realized without any discrepancy. In summary, we have developed a test method for the smoke control capabilities of door sets in the field through testing and theoretical analysis. In the past, the detection method could only be used in the laboratory, and the detection equipment could not be moved. The instrument developed by this research is light and easy to carry, and the operation method is simple. Such a test method can be applied to any doors and is non-destructive, non-hazardous, and reusable, enabling an immediate understanding of the door's smoke control performance in the field. In the future, by extending the design principle of the system, the test method can also be applied to other fire protection equipment for the inspection of smoke control capabilities and can be used as a reference for relevant organizations to establish test specifications and standards. This study offers the designs and descriptions of the equipment utilized with the intention of sharing this information for future reference.

## 2. Experimental Plan

### 2.1. Scope and Conditions

#### 2.1.1. Scope

This test method is suitable for all doors, including single doors, double doors, elevator doors, roll-up doors, etc. It is a verification method for evaluating the smoke control performance of doors at ambient temperature. The prerequisite is that each door has passed the test of CNS 15038 [3].

#### 2.1.2. Conditions

Before the field test, the thermometer, relative humidity meter, atmospheric pressure meter, gas volume flow meter, and differential pressure meter required for the test must be calibrated by a TAF—(Taiwan Accreditation Foundation) accredited laboratory. It must also meet the requirements of Section 2.2.

#### 2.1.3. Terms and Definitions

① 　Door Assembly

A combination of fixed structures (such as doorframes), doors, and attached hardware accessories.

②	Ambient Temperature

The ambient temperature of this test method is (25 ± 15) °C.

③	Leakage

The volume of air through the door assembly under a pressure difference.

### 2.1.4. Leakage Regulations

The volume leakage under a pressure difference of 10 and 25 Pa should be measured separately, with the benchmark of 25 Pa pressure difference, which is converted to the volume leakage under standard conditions that should not be greater than 25 m³/h. In addition, the volume leakage under a pressure difference of 10 Pa should be free of abnormalities.

### *2.2. Experiment Apparatus*

Apart from thermal treatment systems, the concepts utilized in the test method of this study are consistent with CNS 15038 [3], ISO 5925-1 [5], and BS 476-31 [11]. It is mainly based on the use of gas volume flow meters in conjunction with differential pressure meters to measure the volume leakage of test bodies. The configuration of the test equipment is shown in Figure 1, which can be further classified into three main components.

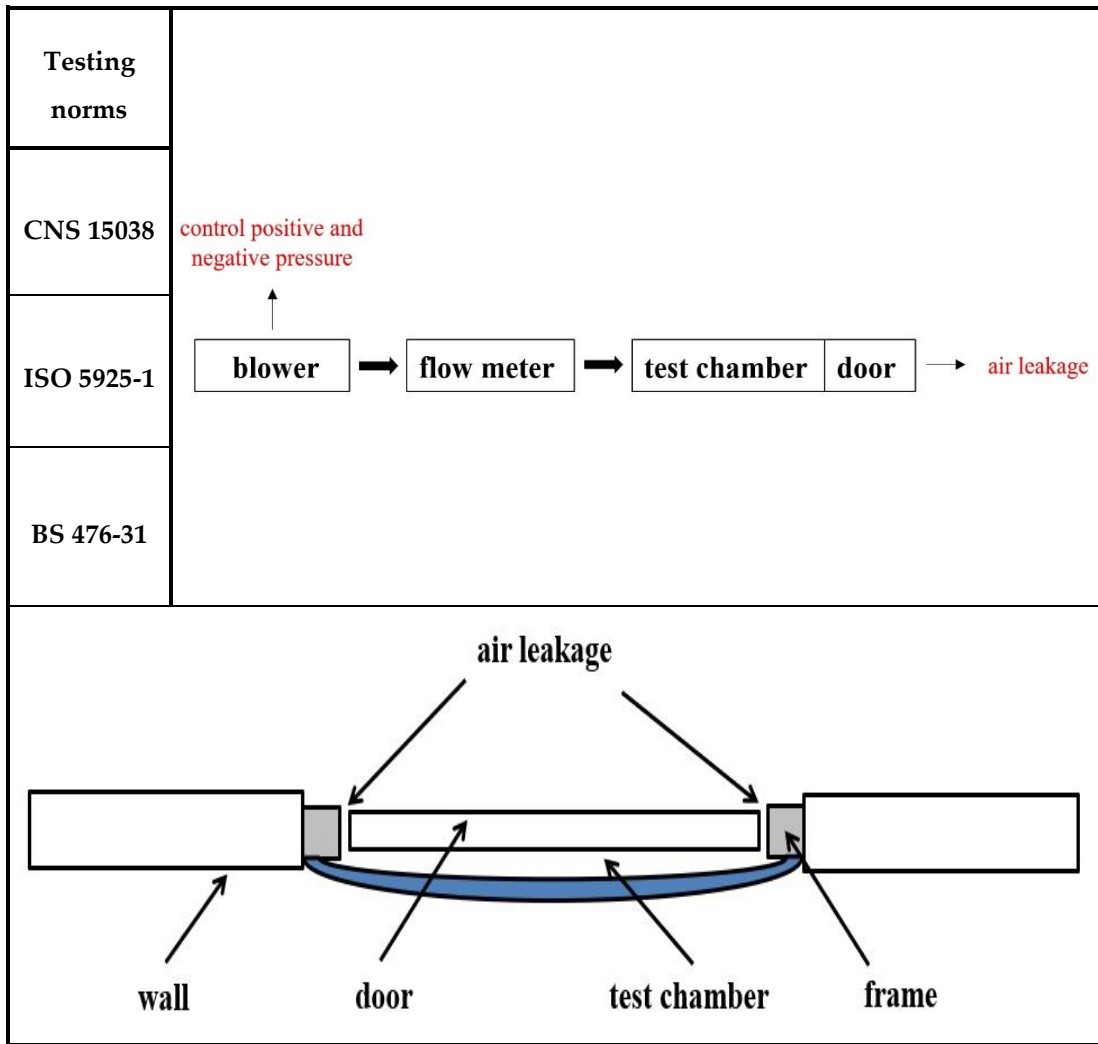

**Figure 1.** Configuration of the test equipment.

Component 1—Test chamber:

The test chamber consists of plastic sheeting and the test body, formed principally by anchoring the plastic sheeting around the door frame with airtight tape.

Component 2—Measurement system:

This includes ① a thermometer, ② a relative humidity meter, ③ an atmospheric pressure meter, ④ a gas volume flow meter, and ⑤ a differential pressure meter.

①　Thermometer measurement range: $-40\,°C \sim +100\,°C$, with an accuracy of $\pm 5\%$.
②　Relative humidity meter measurement range: 0% RH~100% RH, with an accuracy of $\pm 5\%$.
③　Atmospheric pressure meter measurement range: 300 hPa~1200 hPa, with an accuracy of $\pm 5\%$.
④　Gas volume flow meter: Honeywell (Honeywell International, Inc., Charlotte, NC, USA) intelligent differential pressure transmitter with flow meter. The measurement range is $0\ m^3/h \sim 75\ m^3/h$; the accuracy is $\pm 2.5\%$; the applicable fluid temperature is $-10\,°C \sim +60\,°C$; and humidity is below 90%. The flow meter has an inlet and outlet pipe diameter of 50 mm and is installed between the blower outlet and the test chamber.
⑤　Differential pressure meter: Testo 510 (Testo SE & Co. KGaA, Titisee-Neustadt, Germany) pocket-type micro differential pressure meter, used to measure the difference in static pressure between the inside and outside of the test body; its measurement range is 0 hPa~100 hPa; its accuracy is $\pm 0.03$ hPa; and it is placed at $100 \pm 10$ cm from the surface of the test chamber in front.

Component 3—Inflating system:

The blowing engine from Y.H. Industrial Co., Ltd. (Taipei, Taiwan) with adjustable air speed can provide a stable air supply and pressure and is capable of maintaining a uniform pressure difference with a maximum air volume of 6.8 m3/min, 1/4 HP, a voltage of 220 V, three-phase electricity (Three-Phase), and a 50 mm outlet diameter, where the engine frequency control ranges from 0.01 HZ to 650.00 HZ in combination with a Teco Electric & Machinery Co. converter.

*2.3. Test Principles*

The configuration of the equipment used in the field test is shown in Figure 2, with a single test-chamber approach. The test chambers were uniformly positioned in the opening direction of the door (Figure 3) based on the previous results of Chuang et al. [19], which revealed that the volume of air leakage from both sides of the door at ambient temperatures measured was approximately equivalent. Therefore, it can be inferred that, regardless of which direction the door was tested, the results of leakage from the door should be representative. In addition, test chambers were uniformly established in the opening direction of the door for testing, in order to test the functionality of the test bodies through a functional operation of opening and closing each door more than five times [3] prior to starting the test. In the case where the test apparatus had been installed on site, as well as considering the use of positive pressure on the opening direction of the door according to common sense, there is no door frame to support the door, leaving only the automatic door closer and the lock. Therefore, the testing direction of the smoke control field test should be standardized to the open side of the door. This method of inspection should be relatively critical, and the actual volume of leakage after the installation of the door should be more detectable. The functionality of the automatic door closer and the lock can also be tested.

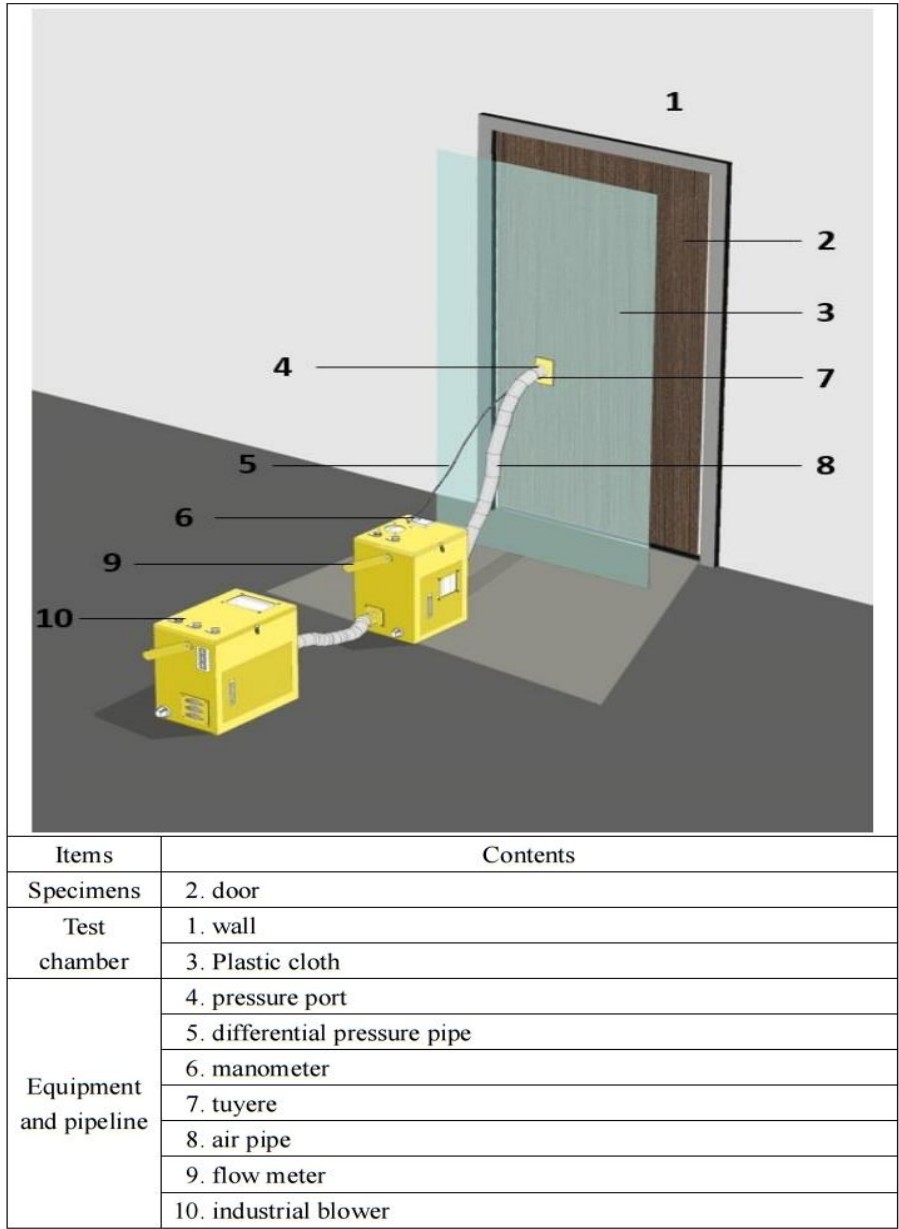

| Items | Contents |
|---|---|
| Specimens | 2. door |
| Test | 1. wall |
| chamber | 3. Plastic cloth |
| Equipment and pipeline | 4. pressure port |
| | 5. differential pressure pipe |
| | 6. manometer |
| | 7. tuyere |
| | 8. air pipe |
| | 9. flow meter |
| | 10. industrial blower |

**Figure 2.** Schematic diagram of the configuration of the equipment for the field test.

This test method was based upon the test chamber design example in Section 3.1 of CNS 15038 [3], the test system composition sketch in ISO 5925-1 [5], and the instrument assembly diagram in the journal published by Liu et al. [20]. In the previous version of CNS 15038 [21], amendments were made in accordance with the previous version of ISO 5925-1 [22] and ISO 5925-2 [23]. The subsequent revisions were also made in the updated version of CNS 15038 [3] in accordance with the updated version of ISO 5925-1 [5] and ISO 5925-2 [6], respectively. Regardless of how the specifications were revised throughout the years [3,5,6,21–23], it was found that with the exception of the medium-temperatures section, which is subject to a change in heating rate, all versions adopted a single test-chamber approach. Therefore, it was considered feasible for this study to utilize a single-chamber approach to measure the volume leakage of the door at ambient temperatures.

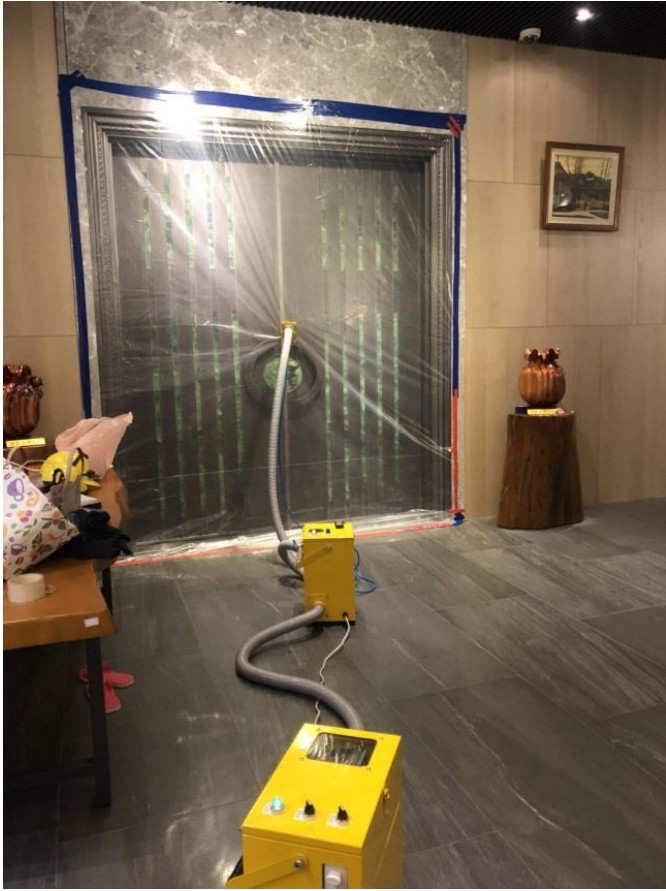

**Figure 3.** Field test images.

*2.4. Test Procedure*

The standard test procedure for the volume leakage of doors is as illustrated below:

Step 1. Pre-test environmental assessment

Measure the atmospheric pressure, temperature, and relative humidity of the test site. The temperature should be 25 ± 15 °C and the relative humidity should be 40–90% before the test is conducted.

Step 2. Door functionality assessment

Each door should be opened and closed (to a minimum of 30 degrees) more than 5 times [3]. In general, the opening direction of the door is the test surface. In the case of circumstances not permitting this, the other side can be used instead if the test cannot be conducted in this direction.

Step 3. Trimming the plastic sheeting

After the plastic sheeting is trimmed to the appropriate size for the door, a hole should be dug in its center for the tuyere and the pressure port.

Step 4. Establishment of the test chamber

Wipe the door frame to be tested with a rag, cover with plastic sheeting, and fix to the outside of the door frame with airtight tape.

Step 5. Equipment installation

Install the air supply tube and differential pressure tube in the tuyere and pressure port, and connect to the gas volume flow meter and differential pressure meter.

Step 6. Power on the equipment

After powering on the test equipment, zero and confirm the differential pressure meter and the gas volume flow meter.

Step 7. Environmental testing

Environmental testing should be conducted continuously for more than 3 min before the measurement of the volume leakage of the test body.

Step 8. Testing and recording

Activate the pressurization system and record the volumetric flow rate. Conduct the leakage measurement in accordance with the setting of 10 and 25 Pa differential pressure between the inside and outside of the test chamber. Under different differential pressure conditions, record every 30 s and calculate the average leakage during the 2 min.

Step 9. Restoration of the field site

Upon test completion, remove all equipment and restore the site to its original state.

## 3. Results and Discussion

### 3.1. Confirmation of Device Function

The precision instruments applied for testing purposes mainly involve a gas volume flow meter, a differential pressure meter, and a blowing engine, and all three of them should be calibrated. Upon completion of calibration, the measured results should be theoretically accurate provided that the equipment is not subjected to impacts. However, since this equipment is used in field tests, it is inevitable that there will be vibrations due to the frequent transportation. Therefore, before each official test, it is necessary to confirm the accuracy of the instrument to maintain the credibility of the test results. As a result, this study designed a test tube to be used for verification prior to official testing (with an inner diameter of 15.24 cm, a length of 100 cm, and a 2 cm diameter round hole in the center of one end of the tube for differential pressure measurement, with the other end connected to an air supply tube). Compare the theoretically calculated volume leakage value with the volume flow value measured by the test tube to confirm the accuracy of the flow meter measurement results. According to theoretical and practical studies in fluid mechanics, the flow coefficient is generally between 0.6 and 0.7 [14]. The different flow coefficients directly affect the calculated value of the flow in the openings. The factors determining the flow coefficients are complex, as the flow coefficients may be influenced by obstructions at the openings, such as the manner in which smoke flows through the openings [24] or the influence of human positioning [25], as well as varying gap widths [13]. For the sake of discussion, the theoretical volume leakage values can be calculated by Bernoulli's equation [26], as shown in Equation (1), assuming a flow coefficient of 0.6 to 0.7, an air temperature of 25 °C, and pressure differences of 10, 25, and 50 Pa.

$$Q = C \times A\sqrt{\frac{2\Delta P}{\rho}} \tag{1}$$

Q: Volume flow rate of air flow through apertures ($m^3$/s);
C: Flow coefficient (C = 0.6~0.7);
A: Flow area or ventilated area ($m^2$/s);
$\Delta P$ Differential pressure between the two sides of the air flow course (Pa);
$\rho$: Density of air entering the flow course (kg/$m^3$), $\rho = \frac{352.8}{273+T_\infty}$;
$T_\infty$: Air temperature (°C).

The calculated leakage values are shown in Table 1, where the volume leakage amounts were 2.79, 4.41, and 6.24 $m^3$/h for pressure differences of 10, 25, and 50 Pa, respectively, at a flow coefficient of 0.6; then, the volume leakage amounts were 3.33, 5.27, and 7.46 $m^3$/h for pressure differences of 10, 25, and 50 Pa, respectively, at a flow coefficient of 0.7. In the future, if the measured volume leakage value is within the theoretical range of 0.60 to

0.70 flow coefficient, the volume leakage measurement equipment can be considered to be good and ready to be applied to the volume leakage measurement of subsequent doors.

**Table 1.** Theoretically projected volume leakage values.

| Flow Coefficient | Differential Pressure | | |
|---|---|---|---|
| | 10 Pa | 25 Pa | 50 Pa |
| 0.60 | 2.79 m³/h | 4.41 m³/h | 6.24 m³/h |
| 0.70 | 3.33 m³/h | 5.27 m³/h | 7.46 m³/h |

*3.2. Flow Test Value Judgment*

Kuo et al. [17] compiled the basic volume leakage requirements of test chambers from various countries. For instance, the previous version of ISO 5925-1 [22] and ISO 5925-2 [23] required the basic volume leakage of test chambers to be less than 1 m³/h, and the previous version of CNS 15038 [21] required the basic volume leakage of test chambers to be less than 2 m³/h. DIN 18095-1 [7] and DIN 18095-2 [8] specified that at a differential pressure of 50 Pa, the basic volume leakage of the test chamber shall not exceed 5 m³/h; BS 476-31 [11] specified that at a differential pressure of 50 Pa, the basic volume leakage of the test chamber shall not exceed 7 m³/h, and the new CNS 15038 [3] specified that the basic leakage of the test chamber shall be less than 7 m³/h. In addition, UL 1784 [9] and JIS A1516 [4] do not specify the requirements of the basic volume leakage of the test chamber. The actual volume leakage of the door test body is defined as the total volume leakage minus the basic volume leakage of the test chamber. From the above, it can be seen that the basic volume leakage of the test chamber does not theoretically affect the actual volume leakage of the door. This is a simple mathematical procedure in which by subtracting the "basic volume leakage of the test chamber" from the "sum of the volume leakage of the test chamber", the volume leakage of the door test body can be found [27], calculated by Equation (2) and converted to the actual volume leakage of the test body in a standard condition.

$$Q_a' = \frac{Q_a}{(T + 273.15)} \times \left[ k \times (p_a + p_m) - 3.795 \times 10^{-3} \times M_w \times p_{H_2O} \right] \qquad (2)$$

$Q_t$: The sum of the volume leakage of the test body and the basic volume leakage of the test chamber (m³/h);

$Q_b$: Basic volume leakage of the test chamber (m³/h);

$Q_a$: Actual volume leakage at temperature (T + 273.15) and pressure (Pa + Pm) of the test body (m³/h), $Q_a = Q_t - Q_b$;

$Q_a'$: Actual volume leakage of the test body under standard conditions (m³/h);

$T$: Air temperature (°C);

$k$: Constant (293.15/101,325) = $2.89 \times 10^{-3}$;

$p_a$: Atmospheric pressure (Pa);

$p_m$: Increased value of pressure (Pa);

$M_w$: Relative humidity (%);

$p_{H_2O}$: Saturation vapor pressure (Pa).

This test method excludes the step of measuring the basic volume leakage of the test chamber due to the field operating environment, which is bound to raise questions regarding the impartiality of this method. However, the basic volume leakage of the test chamber is in the range of 0~0.5 m³/h after repeated tests in the field by taping the contact surface of the plastic sheeting to the door or wall. Theoretically, this method should allow one to more easily achieve airtightness as compared to a chamber consisting of a complex heating system, steel plates, piping, welding, screws, and adhesive strips in a laboratory. Although it is possible to apply adhesives in the field to bring the basic volume leakage of the test chamber to a level closer to 0 m³/h, the basic volume leakage of the test chamber in the field should, for the sake of conservativeness, be 0.5 m³/h for subsequent calculations.

Under different environmental conditions, the measured volume leakage values in the field should be corrected from Equation (2) to Equation (3), as the correction coefficients for different environmental conditions may vary.

$$Q_a' = \frac{Q_t - 0.5}{(T + 273.15)} \times \left[ k \times (p_a + p_m) - 3.795 \times 10^{-3} \times M_w \times p_{H_2O} \right] \quad (3)$$

CNS 15038 [3], ISO 5925-1 [5], DIN 18095-1 [7], and BS 476-31 [11] all specify temperatures in the range of $25 \pm 15\,°C$, i.e., from 10 to 40 °C for ambient-temperatures testing. In CNS 15038 [3], the volume leakage under a pressure difference of 10, 25, and 50 Pa should be measured separately, with the benchmark of 25 Pa pressure difference, which is converted to the volume leakage under standard conditions that should not be greater than 25 m$^3$/h. In addition, the volume leakage under a pressure difference of 10 Pa and 50 Pa should be free of abnormalities. The field test in this study excluded the detection of 50 Pa differential pressure for two reasons, the first of which is based on a comprehensive fire combustion study [28]. The differential pressure in general fires ranges from 10 to 15 Pa, while the field test in this study considered 25 Pa differential pressure as the benchmark, which is already a stringent standard and sufficient to meet the pressure derived from fires. The second is that the test chamber created with plastic tape and plastic sheeting may not be able to sustain the force of 50 Pa differential pressure for prolonged periods of time, which may cause the tape to loosen from the door frame. Although this can be improved by using a greater amount of, and better, tape, the test duration, operating cost, and the cleanliness of the door test body (which affects tape adhesion) are taken into consideration. Under conditions that do not affect the test results, the operation of measuring 50 Pa of differential pressure was excluded from the test results, and the field test was conducted only for the volume leakage measurement of 10 Pa and 25 Pa of differential pressure.

### 3.3. Test Method Verification

For the purpose of verifying the measurement capability of the field test equipment, a steel double flat door (2475 mm (door height) × 2370 mm (door width) × 55 mm (door thickness)) was first tested for volume leakage by this test method (Figure 4), and subsequently the same door was installed in the laboratory [17] for the CNS 15038 [3] volume leakage measurement. The test results were then compared. The temperature was 31.5 °C, with an ambient humidity of 73%, an ambient atmospheric pressure of 100,880 Pa, and a saturation vapor pressure of 4626.6 Pa. Volume leakage measurements were measured 10 times at a differential pressure of 10 Pa and 25 Pa, and recorded at 30, 60, 90, and 120 s during each testing period. The results of each test were then converted into the actual volume leakage of the test body under standard conditions (Figure 5), followed by statistical independent sample testing using SPSS [29] software (Table 2) to analyze the relationship between the two test results.

#### 3.3.1. Levene's Test

The results of the volume leakage were analyzed by F-testing, where the *p*-value = 0.722 > 0.05 for a differential pressure of 10 Pa, indicating no significant difference between the two variables. The *p*-value = 0.718 > 0.05 for a differential pressure of 25 Pa, indicating no significant difference between the two variables.

#### 3.3.2. Independent Samples *t*-Test

The F-test recognized that there was no significant difference in the variance of the volume leakage between the two. Therefore, in the *t*-test, under the condition of equal variances, the calculated t-statistic value of 10 Pa was −1.077, with a two-tailed *p*-value = 0.296 > 0.05. The calculated t-statistic value of 25 Pa was −1.467, with a two-tailed *p*-value = 0.160 > 0.05; therefore, the null hypothesis cannot be rejected.

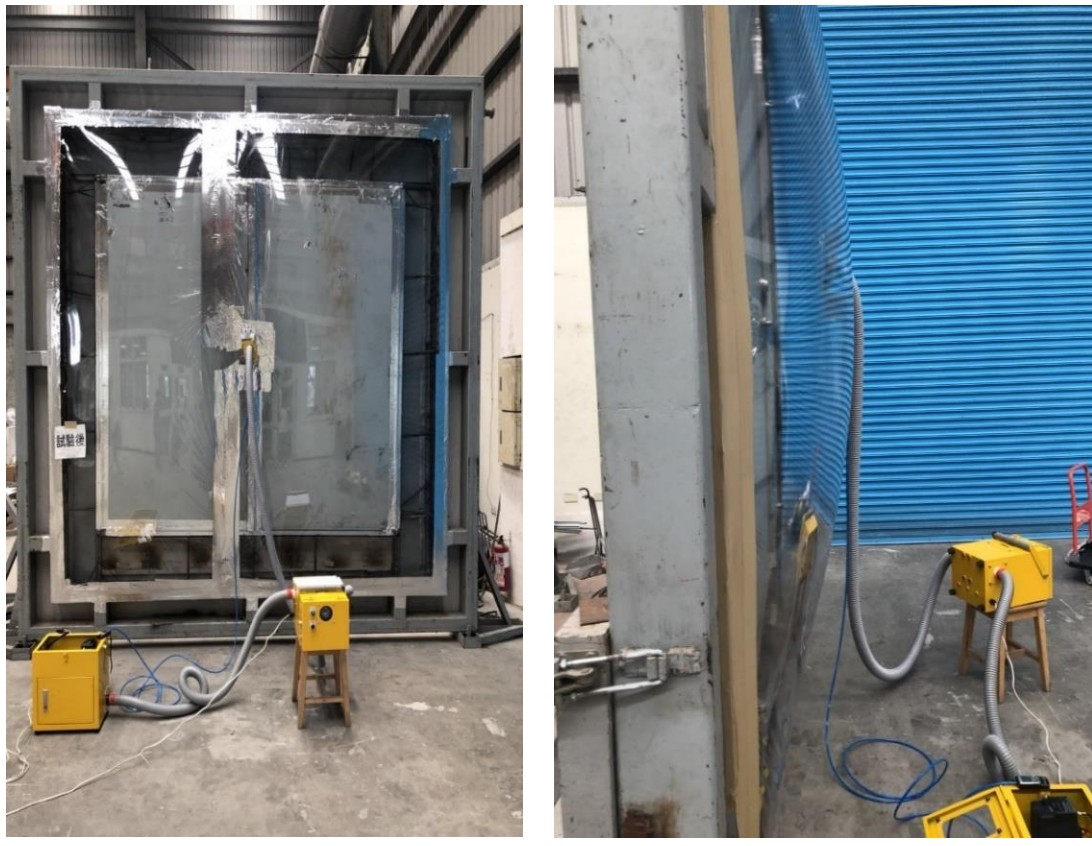

**Figure 4.** Smoke control field test method verification.

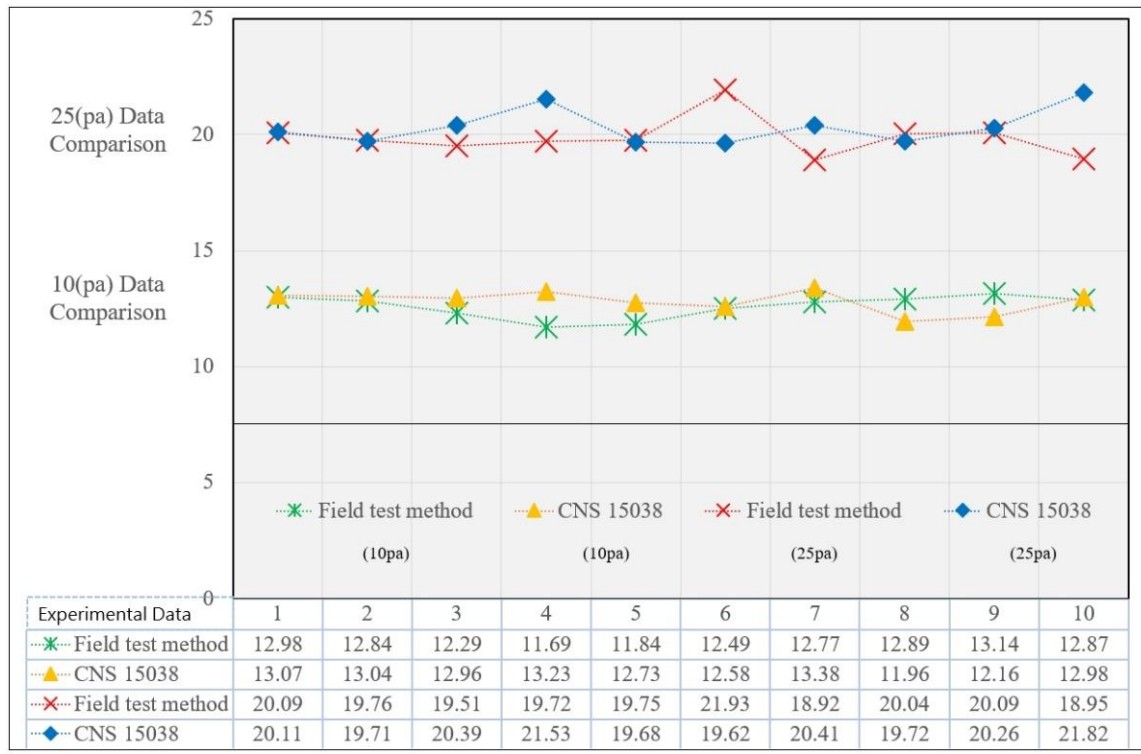

| Experimental Data | 1 | 2 | 3 | 4 | 5 | 6 | 7 | 8 | 9 | 10 |
|---|---|---|---|---|---|---|---|---|---|---|
| Field test method | 12.98 | 12.84 | 12.29 | 11.69 | 11.84 | 12.49 | 12.77 | 12.89 | 13.14 | 12.87 |
| CNS 15038 | 13.07 | 13.04 | 12.96 | 13.23 | 12.73 | 12.58 | 13.38 | 11.96 | 12.16 | 12.98 |
| Field test method | 20.09 | 19.76 | 19.51 | 19.72 | 19.75 | 21.93 | 18.92 | 20.04 | 20.09 | 18.95 |
| CNS 15038 | 20.11 | 19.71 | 20.39 | 21.53 | 19.68 | 19.62 | 20.41 | 19.72 | 20.26 | 21.82 |

**Figure 5.** Smoke control field test method and CNS 15038 test results.

**Table 2.** Independent sample testing.

| Differential Pressure | Testing Methodology | Levene's Test | | Independent Samples *t*-Test | | | | | |
|---|---|---|---|---|---|---|---|---|---|
| | | F | *p*-Value | t | Degree of Freedom | *p*-Value (Two-Tailed) | Mean Difference | SD Difference | CI 95% |
| 10 Pa | Equal Variance | 0.131 | 0.722 | −1.077 | 18 | 0.296 | −0.229 | 0.21257 | Lower Limit −0.6756 Upper Limit 0.2176 |
| | Unequal Variance | | | −1.077 | 17.895 | 0.296 | −0.229 | 0.21257 | Lower Limit −0.67579 Upper Limit 0.21779 |
| 25 Pa | Equal Variance | 0.134 | 0.718 | −1.467 | 18 | 0.160 | −0.536 | 0.36530 | Lower Limit −1.30346 Upper Limit 0.23146 |
| | Unequal Variance | | | −1.467 | 17.958 | 0.160 | −0.536 | 0.36530 | Lower Limit −1.30359 Upper Limit 0.23159 |

### 3.3.3. Conclusions

The results of the independent sample testing and analysis showed no significant difference in the volume leakage test results between the two tests at ambient temperatures under differential pressures of 10 Pa and 25 Pa. This suggests that the data measured by the smoke control field test method and the instrumentation used in this study were accurate and reliable, and can therefore be used for field tests.

### 3.4. Actual Experiment

To investigate the feasibility of the smoke control field test method, the equipment was moved to a construction site to conduct tests on a total of 20 doors (test numbers: A to T). The test time is from summer (average 29 °C) to winter (average 15 °C) in Taiwan, to test whether the equipment still has the ability to measure in the face of temperature, air pressure, and humidity changes. During the testing process, the site environment varied for each test, including room doors in first-floor residences, room doors in long-term care facilities, room doors in nursing facilities, escape doors in tunnels, and safety ladder doors on the fortieth floor. The test doors consisted of wood, plastic, aluminum, and iron, respectively. In each test, the plastic sheeting can be fixed to the door only by tape to form the test chamber. The tape used in the smoke control field test was made of aluminum foil (thickness 0.12 mm, width 14 cm, tensile strength 1020 N/100 mm, elongation at break 10%, and adhesion 51 N/100 mm). The entire test process can be completed by two testing personnel within 30 min. The results of each test were converted into the actual volume leakage of the test body under standard conditions (Figure 6), which can be found to possess specific pattern properties at 10 Pa and 25 Pa. As an example, for the test result of test number A, the actual volume leakage under a pressure difference of 25 Pa was converted into 13.73 m$^3$/h under standard conditions, which can be calculated by Equation (1). The theoretical volume leakage value under 10 Pa pressure difference was 5.64 m$^3$/h, which was only 0.19 m$^3$/h away from the field test result of 5.83 m$^3$/h. With a discrepancy of less than 1 m$^3$/h between the theoretical volume leakage value derived from Bernoulli's equation [26] and the volume leakage value measured by the field test, the test data of this study have been proved to be reliable. Under a pressure difference of 25 Pa, the results of the D, H, O, and Q tests were 30.04 m$^3$/h, 39.68 m$^3$/h, 28.73 m$^3$/h, and 37.42 m$^3$/h. The test results were all larger than 25 m$^3$/h and not in compliance with the standard. It was noted that the reason for the failure could be attributed to the inability to adequately fill the airtight strips due to oversized door seams. In the C, I, and P tests, the gap between

the door and the floor was 1, 2, and 1.9 cm. Although the air supply system was operated at full power, the door was installed with an automatic descending airtight strip that failed to adhere closely to the ground, resulting in an excessive volume of leakage from the test chamber that prevented the pressure in the test chamber from rising. Other doors that passed the test (A, B, E, F, G, J, K, L, M, N, R, S, and T) were observed to have airtight strips set up in accordance with regulations and installed in appropriate positions. The test results further corresponded to the fact that while the doors could pass the CNS 15038 [3] test standard in the laboratory, the doors installed in the field may not necessarily achieve the same performance as in the laboratory. Therefore, it is necessary to investigate the actual smoke control performance of the doors in the field through field tests to help maintain the safety of personnel during evacuation scenarios.

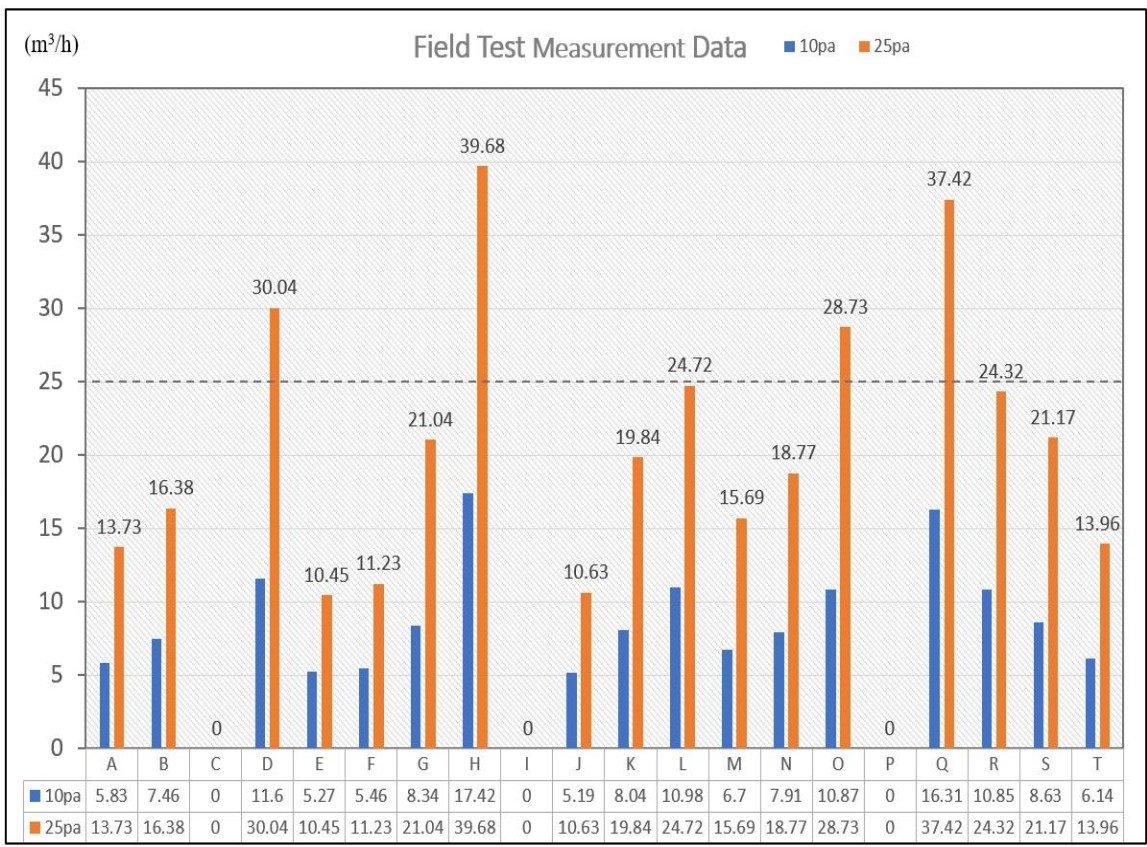

**Figure 6.** Field test measurement data.

## 4. Conclusions

A test tube was designed to allow for verification prior to official testing. Theoretically projected volume leakage values can be compared with the volume flow values measured by the test tube, thereby allowing for an immediate understanding of the accuracy of the measurement results prior to field testing.

Upon analysis, the general differential pressure in the fire scenarios varied from 10 to 15 Pa, considering test duration, operating costs, and the cleanliness conditions of the test bodies (which affects tape adhesion). The operation of measuring 50 Pa differential pressure was excluded under the condition that it does not affect the test results. Therefore, the field test was conducted only for differential pressures of 10 Pa and 25 Pa for the volume leakage measurement.

A comparison test was conducted for this test method and the CNS 15038 [3] method to evaluate the smoke control performance of doors for the same subject, and a statistically independent sample was examined using SPSS [29] software.

The whole test procedure can be completed by two testing personnel within 30 min, without damaging the door or contaminating the environment. It is demonstrated that this method can complete measurements in any environment, and the test results possess certain reliability.

Among the 20 doors tested in the field, only 13 doors passed the field test, thereby demonstrating that the doors installed in the field do not necessarily achieve the same smoke control performance as those in the laboratory, The smoke control performance may deteriorate due to increased gaps between doors and frames or wear and tear of door hinges over time. The above scenarios highlight the need to manage the smoke control performance from the user end in order to ensure the safety of personnel.

Upon many field experiments, the actual leakage is very close to the theoretical values. The volume leakage will increase with enlarged openings or increased pressure differences. In addition to the large leakage rate through the gap beneath the door, other volumetric airflow rates through lockset, hinges, top, and side gaps are very small.

This test method can be directly applied to various doors, including single and double doors, elevator doors, and roll-up doors. In the future, by extending the design principle of the system, the test method can be applied to other fire protection equipment for the inspection of smoke control capabilities, such as building ducts and fire escapes.

**Author Contributions:** Conceptualization, H.-Y.H., C.-Y.L., and Y.-J.C.; data curation, H.-Y.H.; formal analysis, C.-P.L.; Investigation, H.-Y.H.; methodology, H.-Y.H., C.-Y.L., Y.-J.C.; validation, H.-Y.H.; visualization, H.-Y.H.; writing—original draft, H.-Y.H.; writing—review and editing, H.-Y.H. All authors have read and agreed to the published version of the manuscript.

**Funding:** This research received no external funding.

**Institutional Review Board Statement:** Not applicable.

**Informed Consent Statement:** Not applicable.

**Data Availability Statement:** The data are contained within the article. The data presented in this study are available in Figures 5 and 6 and Table 2.

**Conflicts of Interest:** The authors declare no conflict of interest.

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
