# Peer review of "Application Development of Smoke Leakage Test Apparatus for Door Sets in the Field"

_fire, doi:10.3390/fire5010012_

Round 1

Reviewer 1 Report

The paper is generally very interesting and presents original solutions. It seems to me that in the point concerning the analysis of the existing methods and related requirements, the authors should also refer to the EN 1634-3 and EN 13501-2 standards, which are commonly used to assess smoke control doors in the European Union. There are also several English-language publications describing the "European method" that could be used as a references. However, this does not in any way disqualify the paper. Please treat this remark as an opportunity to improve the work, not as necessary to be introduced.

The article should definitely be submitted for proofreading. Some sentences are so long that when you come to their end you forget what was written at the beginning.

Summing up, the only remark is to submit the article for verification by a native speaker. After proofreading, the article will be very valuable and definitely easier to read. 

Reviewer 2 Report

This study is to develop a methodology for the field testing of smoke control properties of doors. this test method has undergone a comparison test with CNS 15038 "Method of test for evaluating smoke control performance of doors" for the same subject, in which the test results showed no significant difference based on independent sample testing, demonstrating the feasibility of this test method and test apparatus. There are some contents can be considered to improve the quality of this manuscript.

1 What is the innovation of this paper, and what is progressiveness of this method compared with previous method?

2 In section 1: The analysis of research status needs to be strengthened.

3 In section 3.1: The flow coefficient is related to the shape of leakage hole, why determined as 0.6-0.7.

4 In section 3.3.2: the sentence “Therefore, the null hypothesis cannot be rejected.” is repeated.

5 The results measured by method proposed in this work is no significant difference with CNS 15038. Why not use CNS 15038 but this proposed test method? Please clarify the advantage of this proposed test method.

6 The measurement error of proposed test method should be analyzed.
